

# Capturing and analyzing pattern diversity: an example using the melanistic spotted patterns of leopard geckos

Tilmann Glimm[1], Maria Kiskowski[2], Nickolas Moreno[3] and Ylenia Chiari[3]

[1] Department of Mathematics, Western Washington University, Bellingham, WA, United States of America
[2] Department of Mathematics and Statistics, University of South Alabama, Mobile, AL, United States of America
[3] Department of Biology, George Mason University, Fairfax, VA, United States of America

## ABSTRACT

Animal color patterns are widely studied in ecology, evolution, and through mathematical modeling. Patterns may vary among distinct body parts such as the head, trunk or tail. As large amounts of photographic data is becoming more easily available, there is a growing need for general quantitative methods for capturing and analyzing the full complexity and details of pattern variation. Detailed information on variation in color pattern elements is necessary to understand how patterns are produced and established during development, and which evolutionary forces may constrain such a variation. Here, we develop an approach to capture and analyze variation in melanistic color pattern elements in leopard geckos. We use this data to study the variation among different body parts of leopard geckos and to draw inferences about their development. We compare patterns using 14 different indices such as the ratio of melanistic versus total area, the ellipticity of spots, and the size of spots and use these to define a composite distance between two patterns. Pattern presence/absence among the different body parts indicates a clear pathway of pattern establishment from the head to the back legs. Together with weak within-individual correlation between leg patterns and main body patterns, this suggests that pattern establishment in the head and tail may be independent from the rest of the body. We found that patterns vary greatest in size and density of the spots among body parts and individuals, but little in their average shapes. We also found a correlation between the melanistic patterns of the two front legs, as well as the two back legs, and also between the head, tail and trunk, especially for the density and size of the spots, but not their shape or inter-spot distance. Our data collection and analysis approach can be applied to other organisms to study variation in color patterns between body parts and to address questions on pattern formation and establishment in animals.

## INTRODUCTION

Animal color patterns vary within and among individuals, including variation among distinct body parts such as the head, trunk, tail, wings, or ventral or dorsal sides, possibly in response to different selection pressures (*Caro, 2005*; *Forsman et al., 2008*; *Allen et al.,*

Corresponding author
Tilmann Glimm, glimmt@wwu.edu

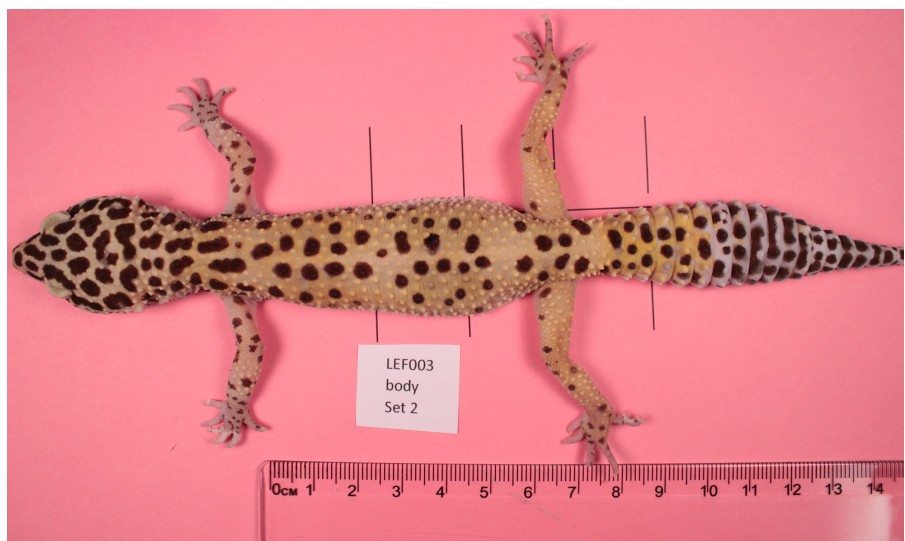

**Figure 1  Example gecko image.** Adult leopard gecko (animal #21003). Note the differences in patterning on the head, trunk, legs and tail.

*2020*). Color patterns may differ in qualitatively obvious ways, such as stripes on the tail and spots on other parts of the body, or in more subtle ways, such as spots of different density or sizes (*e.g.*, Fig. 1). Variation in color pattern is considered a classical example of an adaptive trait, as it is often involved in communication among conspecifics, intrasexual competition, and antipredator functions (*Caro, 2005*; *Gomez, Théry & Losos, 2007*; *Tibbetts & Dale, 2004*; *Solan et al., 2019*).

Although color patterns have been studied extensively because they are involved in many functions essential to the survival and reproduction of organisms, describing and quantifying pattern variation in a multivariate manner is still challenging. Color patterns are generally described in terms of macroscopic differences, such as spots, stripes or labyrinthine organization (*e.g.*, *Miyazawa, Kondo & Okamoto, 2010*; *Allen et al., 2020*; *Kuriyama et al., 2020*), with pattern variation for the same pattern type often quantified by aligning homologous pattern features *via* manually set landmarks or automated image registration algorithms (*e.g.*, *Van Belleghem et al., 2018*; *Bainbridge et al., 2020*; *Prinsloo, Postma & De Bruyn, 2019*) or by focusing on differences in pattern elements, coarsely defined in terms of relative size and position (*e.g.*, *Van den Berg et al., 2020* and references therein). However, complex patterns with high degree of dissimilarity within and among individuals in terms of shape, clustering, size and position of the pattern elements may require the development of new methods to finely capture these differences (see for example *Lee, Cavener & Bond, 2018* and references therein). This is particularly pertinent if the shape and density are irregular and do not fit within specific pattern categories, such as stripes or spots (*Solan et al., 2019*; *Troscianko, Skelhorn & Stevens, 2017*; *Lee, Cavener & Bond, 2018*; *Miyazawa, Kondo & Okamoto, 2010*; *McGuirl, Volkening & Sandstede, 2020*; *Allen et al., 2020*).

Color patterns can be studied using pattern recognition, which broadly speaking deals with classification of image patterns through extraction of significant features (*Zerdoumi et al., 2018*). Methods in this area generally consist of machine learning techniques, that is, prediction systems based on an existing data set. For example, this could entail classification of skin patterns based on a large training data set of skin pattern images. While this approach is certainly viable for synthetic data, *i.e.,* computer-generated patterns (*McGuirl, Volkening & Sandstede, 2020*), where data with thousands of patterns can be amassed easily, this approach may not be feasible for actual images of live animals, where the process of image acquisition is laborious and time consuming. In addition, the use of machine learning techniques would not necessarily provide qualitative insights into what would make two or more patterns similar or different (*Domingos, 2012*; *Zerdoumi et al., 2018*), therefore impeding investigation of what elements of the pattern for example may be more or less variable, constrained or under selection. We note however that powerful machine-learning algorithms for generic image feature extraction such as saliency maps (*Simonyan, Vedaldi & Zisserman, 2014*) may be useful in the binarization of the images. In addition, the results of certain machine learning algorithms can yield human-interpretable results, such as distance scores between patterns, see *Cuthill et al. (2019)*, who classified butterfly wing patterns using a deep learning approach.

Color patterns in leopard gecko skins are formed by pigment-containing cells called chromatophores (*Szydłowski et al., 2020*). There are several types of such chromatophores, classified by the color of their pigments: xanthophores contain yellow pigment, erythrophores contain red or orange pigment and iridophores contain light-reflecting platelets which results in white, blue or red skin coloration. The most easily visible pigment cells are melanophores, dark brown cells containing melanin, which produce the characteristic melanistic patterns of dark spots or stripes (see Fig. 1).

In this article, we address the problem of describing and quantifying variation in melanistic patterns in live geckos *via* computing fourteen different indices, such as the fraction of dark areas to light ones, or the mean size, number and shape of pattern features. Each of these different indices captures only one aspect of the pattern, but collectively, they yield a comprehensive characterization of the pattern itself. Thus, each pattern of the seven body parts studied for each individual corresponds to a point in an abstract 14-dimensional space (here called "pattern space" or "phenotype space"). Our approach is similar to that of *Lee, Cavener & Bond (2018)* who used 11 indices to characterize giraffe coat patterns and *Miyazawa, Kondo & Okamoto (2010)*, who used two indices to describe salmonid fish skin patterns. In contrast to those papers, however, we not only compare single indices between individuals and among individuals, but also consider different ways to measure the overall similarity of two patterns based on their distance in pattern space, taking into account biological information in the data set. In this, our approach is therefore innovative. Arguably, there is no canonical distance function on this multidimensional space to measure overall similarity of patterns, and so we employ two different notions of a metric, both weighted Euclidean distances. These two distances differ in the type of biological information that they may provide. The first distance, the standard Mahalonobis distance, essentially weights each principal component by the inverse of its variance
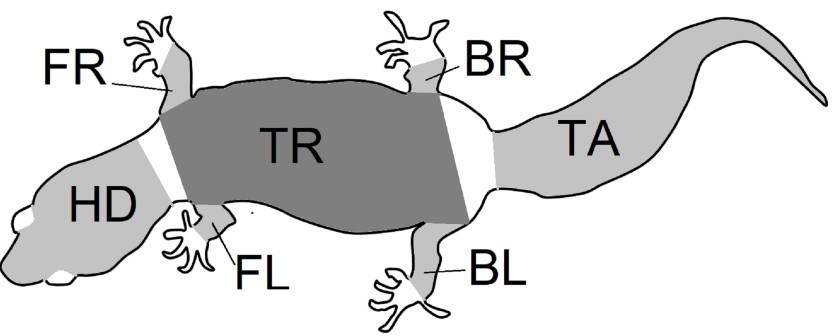

**Figure 2  Body parts.** Outline of the gecko body showing the seven regions of patterns that were isolated from gecko images. Each region was photographed separately, and the limbs were gently stretched during photographing. Nevertheless, depending on the gecko configuration, viewing angles and the region shapes were irregular. An image processing algorithm was used to identify the pattern that was viewable within each region irrespective of the shape. The abbreviations BL (left back leg), BR (right back leg), FL (left front leg), FR (right front leg), HD (head), TR (trunk) and TA (tail) are as in Fig. 3 and are used throughout this article.

(*Mahalanobis, 1927*; *Krzanowski, 2000*). This standard metric weighs all data points equally and thus does not take into account any inherent structure of the data set, as for example any developmental relationship among body parts. The second distance that we selected is instead a measure that weights differences in patterns by the influence of random noise in the developmental process. We call this distance the "Developmental Noise distance". In it, differences in indices for which developmental noise has a small contribution are weighted heavier than differences in indices for which it has a larger contribution. The use of these two distance measures therefore not only permits us to quantitatively describe and statistically test pattern variation, but also helps with understanding the developmental sources - genotypic and environmental, or stochastic - of this variation.

We apply our new approach to capture pattern data and our pattern distance measures to investigate the variation of melanistic skin patterns among distinct body parts for 25 leopard geckos (*Eublepharis macularius*) (Figs. 1, 2 and 3). We use these data to analyze different pattern indices and calculate their correlation to infer the order of pattern formation and establishment across the body and to characterize pattern variation on the different body parts within and among geckos. Finally, by comparing the within-individual left–right variation in leg patterns, which is likely due to developmental noise, to the between-individual variation, which is due to genetic and environmental differences in addition to developmental noise, we quantify the influence of developmental noise on pattern variation for the leopard gecko.

Among vertebrates, lizards have often been used as ideal models to study the evolution of color and color pattern in relationship to other ecological, biological, and behavioral traits (*e.g.*, *Olsson, Stuart-Fox & Ballen, 2013*; *Pérez, DeLanuza & Font, 2016*; *Murali, Merilaita & Kodandaramaiah, 2018*; *Allen et al., 2020*). Specifically, the leopard gecko is an ideal organism on which to study pattern development (*e.g.*, *Chang et al., 2009*). This species is commonly bred in captivity to obtain distinct colors and color patterns, a major advantage

| | Morph | Source | ID | Sex | BL | BR | FL | FR | TR | HD | TA |
|---|---|---|---|---|---|---|---|---|---|---|---|
| 1 | normal | w | 10001 | F | YES | YES | YES | YES | YES | YES | YES |
| 2 | normal | g | 21001 | F | YES | YES | YES | YES | YES | YES | YES |
| 3 | normal | g | 21002 | F | YES | YES | YES | YES | YES | YES | YES |
| 4 | normal | g | 21003 | F | YES | YES | YES | YES | YES | YES | YES |
| 5 | normal | g | 22010 | M | YES | YES | YES | YES | YES | YES | YES |
| 6 | normal | b | 22013 | M | YES | YES | YES | YES | YES | YES | YES |
| 7 | normal | b | 41002 | F | YES | YES | YES | YES | YES | YES | YES |
| 8 | normal | b | 41004 | F | YES | YES | YES | YES | YES | YES | YES |
| 9 | normal | b | 42001 | M | YES | YES | YES | YES | YES | YES | YES |
| 10 | normal | b | 42002 | M | YES | YES | YES | YES | YES | YES | YES |
| 11 | normal | b | 42003 | M | YES | YES | YES | YES | YES | YES | YES |
| 12 | normal | b | 42004 | M | YES | YES | YES | YES | YES | YES | YES |
| 13 | normal | g | 22011 | M | NO | NO | YES | YES | YES | YES | YES |
| 14 | normal | b | 22012 | M | NO | NO | YES | YES | YES | YES | YES |
| 15 | normal | w | 10002 | F | NO | NO | NO | NO | YES | YES | YES |
| 16 | normal | g | 21008 | F | NO | NO | NO | NO | YES | YES | YES |
| 17 | normal | b | 41001 | F | NO | NO | NO | NO | YES | YES | YES |
| 18 | normal | b | 42005 | M | NO | NO | NO | NO | YES | YES | YES |
| 19 | normal | g | 21007 | F | NO | NO | NO | NO | NO | YES | YES |
| 20 | normal | b | 41003 | F | NO | NO | NO | NO | NO | YES | YES |
| 21 | LF | t | 21019 | F | YES | YES | NO | NO | YES | YES | YES |
| 22 | LF | t | 21020 | F | YES | YES | NO | NO | YES | YES | YES |
| 23 | LF | t | 22016 | M | YES | YES | NO | NO | YES | YES | YES |
| 24 | LF | t | 22014 | M | YES | NO | NO | NO | YES | YES | YES |
| 25 | LF | t | 21018 | F | NO | NO | NO | NO | YES | YES | YES |

**Figure 3** **Description of geckos.** Gecko morph ("normal" or "lemon frost" - LF), source, ID, sex (M =male, F =female), and presence of a visible (YES) or not discernible (NO) melanistic pattern on the seven different body parts for each gecko. "BL" and "BR" indicate the back right and left legs, respectively; "FL" and "FR" indicate the frontal left and frontal right legs, respectively. "TR", "TA" and "HD" indicate the dorsal part of the trunk, tail and head of the animal, respectively. (Sources of obtained geckos: w— Greg Watkins—Colwell, g—Tony Gamble, b—Backwater Reptiles, t—Ron Tremper).

when trying to unveil the mechanisms producing variation at these traits (*Cieslak et al., 2011*). Furthermore, our previous work on melanistic patterns on the head of this species (*Kiskowski et al., 2019*) suggests that developmental noise may be an important contributor to its variation.

This work therefore not only proposes a novel approach to analyze similarities in pattern space within and among individuals, but also contributes to our understanding of melanistic pattern formation and establishment in the leopard gecko. The data capture and analysis methods presented here can also be applied to study variation in melanistic pattern elements for developmental, ecological and evolutionary purposes in other organisms. Furthermore, our previous work used mathematical modeling of the process of skin pattern formation to elucidate the influence of developmental noise on patterning (*Kiskowski et al., 2019*). Many different other mathematical models for skin patterning have been proposed (*e.g.*, *Murray, 2002*; *Cruywagen, Maini & Murray, 1992*; *Painter, 2001*; *Cooper et al., 2018*; *Kondo, Iwashita & Yamaguchi, 2009*). In this context, developmental noise can be modeled for example by using random initial conditions for the equations governing pattern formation. Because of this randomness, the resulting pattern is not deterministic, but rather a certain range of possible patterns - all depending on randomization of these initial conditions - may be generated with an associated probability density function. This range

can be thought of as a set of points in the 14-dimensional pattern space provided by the measures used in the current work. This approach and the empirical data can then be used to assess the validity of the models, and thus in turn gain biological insights into the patterning process (see *McGuirl, Volkening & Sandstede, 2020* for an example).

## MATERIALS & METHODS

### Ethical statement

All experiments were carried out in accordance with George Mason University animal use (IACUC) protocol # 1430668.

### Geckos and photographs

For this study we used a total of 25 live adults of *Eublepharis macularius*, the leopard gecko, giving a total of 132 patterns on various body parts. 20 geckos had an overall "normal" pattern morphotype (melanistic/black spots on a yellowish/brownish background), while five geckos had a "lemon frost" morphotype with melanistic patterns (*Szydłowski et al., 2020*; *Guo et al., 2020*. See Fig. A1 for full body images of all geckos, Fig. 3 and Supplemental Information for details on the origin of the geckos).

We photographed the geckos one at the time by placing each one of them on a smooth surface covered with colored paper chosen to contrast well with the full set of geckos in at least one color channel (Fig. SA1). Perpendicular reference lines were printed on the colored paper to ensure placement of the gecko in the same position across picture sets. For each gecko, we obtained four picture sets to measure the error associated with the data capture. (See Supplemental Information for further details.)

### Image analysis

The melanistic spotted patterns were studied on the head, four limbs, dorsal trunk, and tail of each of the 25 geckos. There were thus $25 \times 7 = 175$ separate body parts analyzed in this work (Fig. 4). Not all these body parts showed melanistic spotted skin patterns and only twelve of the geckos had qualifying spotted patterns on all seven body parts (Fig. 3; see details below for how qualifying patterns were recognized).

For our analyses, each of the seven regions for each gecko was isolated by automated removal of background pixels from the images where possible and additionally by hand using the GIMP photo editing tool (Fig. 4). When cutting the regions, we worked along the natural boundary of the body and had defined rules for the edges of the body region (*e.g.*, the trunk was separated from the head by a straight line segment connecting the two most anterior points where each front leg met the main body, and the legs were separated from the body using a straight line segment perpendicular to the limb that was the most proximal line segment that could be drawn without including any portion of the trunk. See Fig. SA3 for more details). We note that the use of these rules meant that only a subset of morphological regions was selected from the entire body. For example, the cervical and sacral regions were not included in either the head region nor the trunk region due to a lack of clear landmarks in this region to define the boundary. For each of the 700 images (25 geckos, seven body parts, four independent photos of each), we identified and isolated the

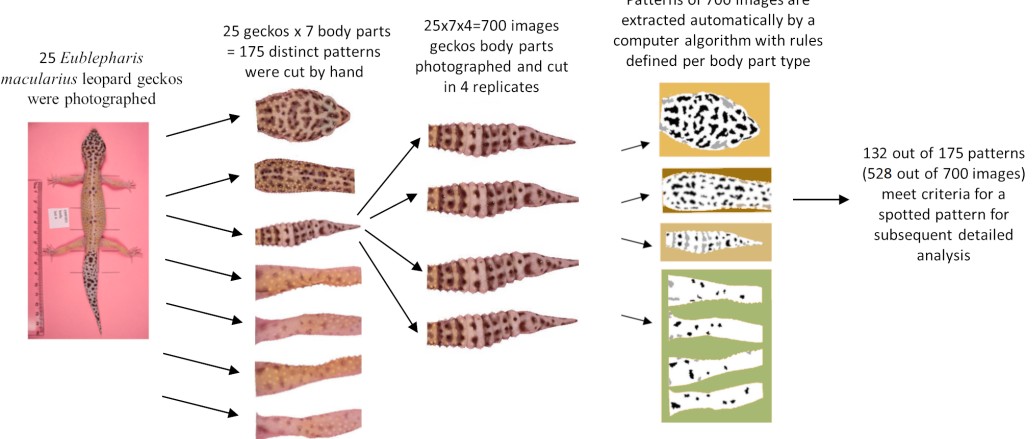

**Figure 4** **Flow chart.** Flow chart of image acquisition and processing.

**Table 1** **Determination of thresholds.** Algorithm criteria for determining the threshold for spots for the four body parts (limbs, head, trunk, and tail) including the color channel that was used for identifying spots (the green channel, for all) and the color channel that was used for identifying regions of the picture requiring lighting adjustment (the green channel for the legs and the blue channel for the trunk; the mean and standard deviation of the pixel values in the region are denoted as $\mu_G, \sigma_G$ and $\mu_B, \sigma_B$, respectively) and the final threshold rule that was applied to the green channel.

| Body part type | Channel used to compute the mean pixel intensity $\mu$ | Channel and threshold used to identify shadows | Threshold |
|---|---|---|---|
| Limbs | Green | shadows: green $< \mu_G - 0.3\sigma_G$ <br> glare: green $> \mu_G + 0.3\sigma_G$ | maximum of these two quantities: $\mu_G - 0.85\sigma_G, 60$ |
| Head | Green | N/A | maximum of these two quantities: $\mu_G - 0.50\sigma_G, 60$ |
| Trunk | Green | shadows: blue $< \mu_B - 0.85\sigma_B$ | maximum of these two quantities: $\mu_G - 0.85\sigma_G, 60$ additionally, maximally: 108 |
| Tail | Green | N/A | maximum of these two quantities: $\mu_G - 0.85\sigma_G, 60$ |

spotted melanistic pattern as a simple binary pattern of black pixels on a white background (*i.e.,* in every image, each pixel has either value of 1 = black or 0 = white). Using criteria to help in isolating the melanistic spotted pattern amongst other patterns of the skin and background noise of the image, a threshold was applied to each of the regions to define the binary pattern of black spots on a white background (see below for details and Table 1). A spot identification algorithm was applied for each type of body pattern to identify spots. A final image processing step with Matlab removed stray pixels, filled in holes, and smoothed the contour of the spots to generate the final spot patterns used for measuring the pattern statistics. The Matlab code for these scripts is available in an Open Science Framework (OSF) repository. Link: https://osf.io/zauwe/?view_only=97c4846fb5f34357b0943c2f5c6bc571.

## Limb, trunk, head, and tail spot identification algorithms

We designed an algorithm to identify the melanistic pigmentation patterns for the four types of body parts (legs, trunk, head, tail) of the 25 geckos. Melanistic patterns were
typically spotted, but there were occasionally stripes or more complex shapes for some body parts and for some geckos.

To identify these patterns, we first binarized the images by labeling each pixel by a value of either 1 or 0, corresponding to relatively darker and lighter regions, respectively. A straightforward method for identifying relatively darker regions - and thus to carry out the binarization step - is to identify pixels that are darker than the average pixel brightness by a fixed multiple of the standard deviation. This approach is popular in cell imaging and document imaging applications as ''robust background thresholding'' and found within the CellProfiler pipeline (*Kamentsky et al., 2011*) and as the backbone of the Niblack method (*Niblack, 1986*). This method has the advantage of having a relatively simple rule for thresholding but yet has the flexibility to accommodate patterns with varying contrast. This flexibility is achieved without observer input since the variation of the pixel intensities within the image itself is used to define what is meant by 'relatively' dark. Among our geckos, the melanistic patterns were consistently darker than the background, but the pixel darkness of the melanistic regions, and the absolute difference in darkness between the two binary regions, varied widely among body parts and among geckos.

In a preliminary ''supervised'' step for identifying/defining patterning, rules for the thresholding algorithm were chosen for a good fit between paucity (to minimize the number of criteria) and robust pattern capturing ability to capture the melanistic patterning for the greatest number of gecko images. For example, it was determined during this step that the relatively simple robust background thresholding/ Niblack method (*Niblack, 1986*) was adequate for the given collection of patterns. Due to morphological differences in the four types of body parts (legs, trunk, head, tail), the thresholding was independently supervised for each type of body part for whether the lighting across the images would be adjusted to correct for shadows (since the body parts had different contours) and to choose Niblack's factor for the thresholding (since the body parts had different fractional areas for the patterning). While the algorithms used to determine the melanistic pattern from the photographic images thus varied for shadow removal and with the Niblack factor for each body part, the algorithm was otherwise applied the same way to every gecko image for uniformity and consistency.

All of the specific rules for the algorithm are summarized in the Supplemental Information, see also the MATLAB code in the Open Science Framework (OSF) repository. (Link: https://osf.io/zauwe/?view_only=97c4846fb5f34357b0943c2f5c6bc571.) See Supplemental Information for images of all 7 body parts and binarized images for 25 geckos (times 4 repeated measurements); also see Fig. SA2 for a representative example of the binarized images for one body part for one gecko.

## Image length scale

For each image, the length scaling factor (length per pixel) was computed *via* determining the number of pixels for one centimeter using the imaging software GIMP. This was used in converting values measured in pixels to lengths, see Table 2. For example, 350 pixels (the lower bound for a region of dark pixels to be identified as a spot) was approximately 0.5 mm$^2$. The total areas of the regions evaluated for patterning (for the four types of body

**Table 2  Indices used in this work to describe patterns.** Like *Miyazawa, Kondo & Okamoto (2010)*, we include a measure for the ratio of black to white pixels (FM) and a measure of the circularity of spots (EE) among the 14 variables used in this work. See Supplemental Information for definition of interior spots.

| Abbr. | Name | Units | Description | Spot type used for the computation |
|---|---|---|---|---|
| FM | fraction of melanistic area | dimensionless | Ratio of black pixels to all pixels in binarized image (ratio of melanistic area to total area) | All Spots |
| SS | mean spot diameter ('spot size') | cm | Mean of the length of the major axes of the ellipses that have the same second moment as the spots (Matlab function *MajorAxisLength*) | Interior Spots |
| SSD | st.dev. of SS | cm | Standard deviation of the lengths of major axes used in definition of SS | Interior Spots |
| EE | mean ellipticity | dimensionless | Mean ratio of major axes to minor axes of ellipses that have the same second moment as the spots (Matlab function *Eccentricity*) | Interior Spots |
| EED | st. dev. of EE | dimensionless | Standard deviation of the ratios used in definition of EE | Interior Spots |
| PL | peak length | cm | Measure of characteristic wavelength; (typical distance between spots; see *Miura, Komori & Shiota (2000)*) | All Spots |
| MD | mean minimum distance | cm | Mean of the mean distance of the centroid of a spot to the closest three other spots' centroids | All Spots |
| MDD | minimum distance st. dev. | cm | Standard deviation of the distances used in the definition of MD | All Spots |
| SA | spot area | cm$^2$ | Mean area of spots | Interior Spots |
| SAD | st. dev. of spot area | cm$^2$ | Standard deviation of the spot areas | Interior Spots |
| SI | spot intensity | dimensionless | Mean green values (G) of the RGB values of spots (between 0 and 255 for each pixel) | All Spots |
| SID | st. dev. of spot intensity | dimensionless | Standard deviation of the intensities used in the definition of SI | All Spots |
| EL | mean spot elongation | dimensionless | Mean of the spot elongation $EL = \frac{area}{2d^2}$ where $d$ is the thickness of the spot (number of erosion steps needed before the pot disappears) | Interior Spots |
| ELD | st. dev. of spot elongation | dimensionless | Standard deviation of the spot elongation | Interior Spots |

parts) varied from 125 to 1,882 mm$^2$ (the legs were the smallest patterning regions, varying from 125 to 300 mm$^2$; the tail regions varied from 524 to 1,230 mm$^2$, the head regions varied from 694 to 1,046 mm$^2$ and the trunk regions varied from 953 to 1,882 mm$^2$).

## Application of a threshold to define spots

A straightforward method for identifying relatively darker regions is to identify pixels that are darker than the average pixel brightness by a fixed multiple of the standard deviation.

Generally, for a region of pixels with mean intensity $\mu$ and standard deviation $\sigma$, a central pixel is identified as "dark" (and assigned the value 1) if it is darker than a threshold value $T = \mu - k\sigma$ where $k$ is Niblack's factor (*Niblack, 1986*). To apply the method, multichannel images are converted to a single-channel image by converting a color image to grayscale or selecting a particular channel. Then, in a supervised approach, an initial value of the Niblack parameter $k$ is chosen by the user and then tuned manually for an image or set of images by trial and error to visually optimize spot detection.

This robust background thresholding method /Niblack method was applied with a single algorithm to identify melanistic patterning for all our gecko images. For all thresholding, the Niblack window was the entire region being analyzed (one body part of one gecko). Each pixel of our photographic images was multichannel with an R, a G and a B value, which are integers that each range from 0 to 255. We used the green channel G as a measure of the intensity of a spot since in our images shadows *versus* pigmented regions with similar brightness were best distinguished by differences in the intensity of their green hue (see Table 1). For the trunk, shadows were distinguished by relatively low values in the blue channel. The thresholding was supervised independently for each morphological region (limbs, tail, trunk and head) for the choice of the Niblack factor since the brightness and relative area of pigmented regions varied for each morphological region. For the legs, trunk and tail, a threshold of $\mu - 0.85\sigma$ was chosen to best capture the melanistic pattern across the entire set of geckos. For the head, which had a larger fractional area of melanistic spots, a threshold $\mu - 0.5\sigma$ was determined to identify the set of melanistic spot pixels best. A high fraction of melanistic area relative to total skin area (combined with exceptionally dark spots) would decrease the average intensity to values that were too low to identify more lightly melanistic spots so a rule for a minimum threshold was applied that the threshold for the value of a pixel would be at least 60 out of 255. Pixels with values this low were invariably melanistic. A maximal threshold of 108 out of 255 was applied to the trunk since a very low fractional melanistic area could cause the average intensity to be too large for good spot detection; however, higher threshold values were frequently appropriate for other body parts (for the legs for example) so no maximal threshold was applied for parts other than the trunk.

## Generalization of the method for extracting patterns for other contexts

We found that each morphological region of the gecko (limb, tail, trunk, and head) was best served by a slightly modified pattern extraction algorithm parameters even though in each case we desired to extract the same pattern of interest, a binary pattern corresponding to relatively light *versus* relatively dark regions of pigmentation. This was because there were systematic differences among the morphological regions including differences in background colors of the skin and non-pattern artifacts such as shadows and dimples. Likewise, morphologically tailored algorithms would likely be required for different animal species and for different patterning contexts. For instance, patterns in different colors (*e.g.*, red spots) could be detected *via* an appropriate choice of color channels or linear combination of channels. However, we recommend that even when comparing patterns from a wide variety of sources, ideally a single pattern recognition method should be applied and potential anthropomorphic bias should be minimized.

## Final pattern processing

The application of the threshold identified a set of pixels that are darker than the threshold value for each evaluated image. This set of pixels included stray pixels as well as larger contiguous areas of pixels that were likely to be pigmented spots. A minimum spot size of 350 pixels ($\sim 0.5$ mm$^2$) was required for the spot to be qualified as such in this work. This

number was determined by visual inspection to remove stray pixels and very small dark artifacts. Although the size of pigmented regions varied (especially the size of these regions could be very large on the head and the trunk), none of the pigmented areas that should be identified as spots were smaller than about 750 pixels in total area, so the minimum spot size was chosen to be smaller than the pigmented areas the algorithm needed to identify but larger than most artifacts.

In the last processing steps, any holes within pigmented regions (see Fig. S1F) were filled, and the contour of the spots was smoothed using successive dilations and erosions (this removes small scale granular effects at the edges of spots without changing the shape of the spot). See the Supplemental Information for details.

## Final pattern classification

For both the limb and trunk patterns, the spotted pigmented pattern would occasionally be very faint and barely present or not present at all. To distinguish among these, an image was classified as a patterned only if there were at least four interior spots for limb patterns and at least six interior spots for the trunk and tail. The head was invariably well-patterned, with at least 26 spots found on the heads of all the geckos, so no minimum was applied.

## Description of indices

For each of the 25 geckos, we excluded images without patterns as determined by the algorithm described above (Fig.4 and Table 1) giving a total of 132 qualifying patterns (14–16 patterns for each of the 4 legs, 23 trunk patterns, 25 head patterns and 25 tail patterns). For each such qualifying combination of body parts and gecko, we used Matlab to calculate the 14 indices summarized in Table 2. The value of each index is the average of the 4 independent measurements, giving a total of $132 \times 4 = 528$ images that were analyzed. For an assessment of the measurement error, see below and the Supplemental Information.

## Definition of distances on pattern space: Mahalanobis and developmental noise distances

Each qualifying pattern (the average of the four images of one of seven body parts of one of 25 geckos) is described by the 14 indices in Table 2. Thus we can think of a pattern as a point $\vec{x} = (x_1, \ldots, x_{14})^T$ in a 14-dimensional pattern space. (The superscript "T" denotes the transpose). To quantify the similarity of two patterns, we measure the distance between two points in this pattern space. We consider two different definitions of a distance (metric) on pattern space. Let $\vec{x} = (x_1, \ldots, x_{14})^T$ and $\vec{y} = (y_1, \ldots, y_{14})^T$ denote two points. The first distance we consider is the standard Mahalanobis distance given by

$$d_N(\vec{x}, \vec{y}) = \sqrt{(\vec{x} - \vec{y})^T S^{-1} (\vec{x} - \vec{y})}.$$

Here $S$ is the covariance matrix of the complete data set. One can think of the Mahalanobis distance as the Euclidean distance computed after transforming the data to principal components and normalizing each principal component (*Krzanowski, 2000*). The advantage of this method over the Euclidean distance on the untransformed pattern space is that the principal components are uncorrelated, and so the Mahalanobis

distance is not skewed by correlations between the different indices. It is generally regarded as an appropriate generic choice for a statistical distance in sample spaces with differential variances and correlations (*Krzanowski, 2000*). As a generic choice however, the Mahalanobis distance does not take into account any specific information from the particular structure of our data set. In the case at hand, the data points can be grouped by animal, by body part, or for instance by pairs of front legs or back legs of the same animal. In general, differences among patterns can be attributed to differences in the genotype, the environment experienced, and developmental noise. The differences among patterns within the pairs of data points describing the front legs or the back legs of the same animal are likely primarily the result of developmental noise, as opposed to two different genotypes or different environmental conditions. Data obtained from the legs allow then to separate the influence of developmental noise from genetic and environmental factors. We take this into account and define a second metric called "Developmental Noise metric", defined as a weighted Euclidean metric:

$$d_D(\vec{x},\vec{y}) = \sqrt{\sum_{i=1}^{14} w_i (x_i - y_i)^2},$$

where the weights $w_i$, i $=1$ ,..., 14 are defined *via* the variance of the two front leg patterns of each gecko. More specifically, for the index $i$ (with ($1 \le i \le 14$), let $S_i^n$ denote the variance of the indices of the two front leg patterns of the $n$th gecko (we only included geckos that have patterns on all four legs; see Fig. 3). Then we define the weight $w_i$ as the inverse of the mean of the variances $S_i^n$, i.e.,

$$w_i = \frac{1}{mean(S_i^n)} \quad (1 \le i \le 14),$$

where the mean is taken over all geckos who have patterns on front legs. Note that this distance function is scale invariant, *i.e.*, independent of the units used for the different indices. The reason for using these weights is that the mean variance between the two front legs is a rough measure for the importance of noise in the establishment of the pattern; thus effectively the smaller influence noise has on a measurement, the more weight it is given in the computation of the distance between patterns. While this distance takes into account the special structure of the data set, its computation is based only on a subset of the data, namely leg patterns of those geckos that have patterned front legs. This is in contrast to the standard Mahalanobis metric, which ignores the special structure, but is based on all data points. It is a priori not clear which of these distances is more appropriate, and for this reason we use both in the following analyses. In fact, we found that in general, the results for these two metrics agree qualitatively, giving added confidence in our results (see Results section). Many other reasonable concepts of distances on pattern space are possible. Results are reported in all cases for the *squares* of the distances.

## Quantification of measurement error

We took four independent photos of each body part, where the animal was picked up and rearranged for each repetition so that the four measurements would be independent. A

measurement error was introduced by slight differences in the rotation and placement, especially for the limbs and tail. To quantify the measurement error, we took two approaches: in the first, we compared the mean distances in pattern space between the different measurements to the mean between-individual distances of the same body part. The second consisted of a two-way ANOVA test for the front legs and the back legs, where the two factors are "sides" ($S$, fixed) and "individuals" ($I$, random). For more details, see the "Results" section below.

### Statistical analysis and software

Images of various body parts were extracted *via* automated removal of background pixels and consequent manual selection from photos of the geckos with the image editor GIMP. All computations for image analysis and statistical analysis were performed with Matlab. The computations of statistical significance of results ($p$-values) were performed either with standard statistical tests as implemented in Matlab where indicated, or *via* nonparametric permutation tests (see the Appendix for details on the procedure). The Matlab code for these scripts is available in an Open Science Framework (OSF) repository. Link: https://osf.io/zauwe/?view_only=97c4846fb5f34357b0943c2f5c6bc571.

## RESULTS

### Patterning

Figure 3 shows that not all geckos had melanistic patterns that were identified on all body parts. In some cases there was no visible spot pattern by eye as well, in some cases the pigmented melanistic pattern visible by eye was very light and not discerned by the algorithm.To ensure a robust pattern with enough spots to measure average characteristics, we included a spot pattern only if the algorithm identified a minimum number of spots (at least four or six interior spots). All of the geckos had patterning on the heads and tails, only two were missing patterns on the trunk, and approximately half (13/25) of the geckos were missing patterns on their legs. There is furthermore an identifiable hierarchy of patterning {head;tail} → {trunk} → {front legs;back legs} for each individual gecko, where absence of patterns in one of the body parts entails absence of patterning in all "downstream" body parts with respect to this hierarchy. For instance, absence of patterning on the trunk means that all legs have no patterns as well. There was no clear hierarchy between the front legs and back legs. Of those missing patterns on legs, most geckos (7/13) were missing patterns on both sets of legs, two were missing patterns on their back legs only, and four were missing patterns on their front legs only. Although our sample size is limited for the "lemon frost" morph, there appeared to be an effect of morphotype: for the "normal" morphotype, absence of patterns on the front legs always meant that the back legs were also unpatterned, whereas this was reversed for this morph, since no gecko had patterned front legs.

### Measurement error

We took four independent photos of each body part. To estimate the amount of measurement error in each of the 14 indices, we took two different approaches (see Methods).

In the first, we consider body part patterns as points in 14-dimensional phenotype space and measure the distances between them. We determined first the mean distance of the four repeated measurements of the same body part of the same animal from their centroid. This is an absolute measure of the measurement error. Table SA1 lists these errors for all seven body parts as well as the relative error as the ratio of these errors relative to (i) the mean *between-individual* distances for the same body part, or (ii) the corresponding *within-individual* distances for front or back legs (comparing the left and right leg of the same gecko). The results for both the Mahalanobis distance and the Developmental Noise distance are listed. In all cases, the mean within- or between-individual distances were significantly greater than the mean distance due to the measurement error, with factors varying from 2.2 (back leg, within-individual distance relative to measurement error, Mahalanobis distance) to 86.1 (tail, between-individual distance relative to measurement error, Developmental Noise distance). A factor of 1 would mean that within- or between-individual distances were of the same magnitude (and thus indistinguishable) from measurement error. However these distances were at least twice as large (and often much larger), indicating that the measurement error is relatively small.

In the second approach for characterizing the measurement error, we conducted a two-way ANOVA test where the two factors are "sides" (S; fixed) and "individuals" (I; random) separately for both pairs of front legs and pairs of back legs for each of the 14 indices. This is a standard approach to compare the relative contributions of nondirectional asymmetry (biological) and measurement error (technical variation) in the investigation of paired structures (*Palmer & Strobeck, 1986*; *Merila & Biorklund, 1995*; *Breuker, Patterson & Klingenberg, 2006*). For each index, an F-test yielded that nondirectional asymmetry is making a significant contribution to the variation observed relative to measurement error. The F-values had a median value of 6.8, meaning that the measurement error made up about 15% (median value) of the total observed variation between the left and right leg patterns. See Table SA2 for details.

## General pattern variation across geckos and body parts

For each body part of each of the 25 studied animals, we determined the value of each of the 14 indices listed in Table 2 *via* the mean of four repeated independent measurements.

An examination of the coefficient of variation (ratio of standard deviation and mean) for the 14 indices shows that melanistic pattern is highly variable for measures that concern the proportion of melanistic areas (FM), how large these individual areas are (SA and SSD), and, to a lesser extent, what their typical distances are from each other (PL and MD) (Table A4). This is not an artifact of the measurement error, which actually tended to be larger for MD than the other indices by some measures (see Table SA2). In essence, spots can be in higher or lower density across geckos and body parts and larger or smaller. However, once a melanistic pattern is established, the spots are all similar in shape (EE, EL). Figure 5 displays the pairwise Pearson correlation coefficients showing how much one index is correlated with another. For example, measures of the size of spots such as FM, SS and SA are strongly positively correlated. The two indices of the typical wavelength (roughly representing the typical distance among melanistic areas), PL and MD, are also positively correlated. It is

|     | FM | SS | SSD | EE | EED | PL | MD | MDD | SA | SAD | SI | SID | EL | ELD |
|-----|-----|-----|-----|-----|-----|-----|-----|-----|-----|-----|-----|-----|-----|-----|
| FM | 1 | 0.76**** | 0.7**** | 0.39**** | 0.14 | -0.63**** | -0.5**** | -0.65**** | 0.68**** | 0.62**** | -0.76**** | -0.55**** | 0.42**** | 0.45**** |
| SS | 0.76**** | 1 | 0.9**** | 0.42**** | 0.16 | -0.39**** | -0.063 | -0.27** | 0.97**** | 0.91**** | -0.65**** | -0.35**** | 0.57**** | 0.54**** |
| SSD | 0.7**** | 0.9**** | 1 | 0.38**** | 0.27** | -0.4**** | -0.15 | -0.28** | 0.84**** | 0.93**** | -0.55**** | -0.36**** | 0.6**** | 0.64**** |
| EE | 0.39**** | 0.42**** | 0.38**** | 1 | -0.47**** | -0.4**** | -0.061 | -0.14 | 0.26** | 0.29*** | -0.094 | -0.26** | 0.58**** | 0.37**** |
| EED | 0.14 | 0.16 | 0.27** | -0.47**** | 1 | -0.004 | -0.1 | -0.097 | 0.18* | 0.19* | -0.2* | -0.079 | -0.0041 | 0.15 |
| PL | -0.63**** | -0.39**** | -0.4**** | -0.4**** | -0.004 | 1 | 0.63**** | 0.58**** | -0.33*** | -0.34**** | 0.35**** | 0.35**** | -0.29*** | -0.3*** |
| MD | -0.5**** | -0.063 | -0.15 | -0.061 | -0.1 | 0.63**** | 1 | 0.85**** | -0.029 | -0.06 | 0.28*** | 0.36**** | 0.12 | -0.011 |
| MDD | -0.65**** | -0.27** | -0.28** | -0.14 | -0.097 | 0.58**** | 0.85**** | 1 | -0.23** | -0.21* | 0.49**** | 0.47**** | 0.082 | 0.022 |
| SA | 0.68**** | 0.97**** | 0.84**** | 0.26** | 0.18* | -0.33*** | -0.029 | -0.23** | 1 | 0.93**** | -0.64**** | -0.29*** | 0.45**** | 0.44**** |
| SAD | 0.62**** | 0.91**** | 0.93**** | 0.29*** | 0.19* | -0.34**** | -0.06 | -0.21* | 0.93**** | 1 | -0.58**** | -0.3*** | 0.52**** | 0.54**** |
| SI | -0.76**** | -0.65**** | -0.55**** | -0.094 | -0.2* | 0.35**** | 0.28*** | 0.49**** | -0.64**** | -0.58**** | 1 | 0.46**** | -0.11 | -0.18* |
| SID | -0.55**** | -0.35**** | -0.36**** | -0.26** | -0.079 | 0.35**** | 0.36**** | 0.47**** | -0.29*** | -0.3*** | 0.46**** | 1 | -0.11 | -0.068 |
| EL | 0.42**** | 0.57**** | 0.6**** | 0.58**** | -0.0041 | -0.29*** | 0.12 | 0.082 | 0.45**** | 0.52**** | -0.11 | -0.11 | 1 | 0.9**** |
| ELD | 0.45**** | 0.54**** | 0.64**** | 0.37**** | 0.15 | -0.3*** | -0.011 | 0.022 | 0.44**** | 0.54**** | -0.18* | -0.068 | 0.9**** | 1 |

**Figure 5** **Correlation matrix.** Values color coded from negative (blue) to positive (red). Symbols for each of the 14 indices are as in Table 2. One star (*) indicates p-values less than 0.05, **p-values less than 0.01, ***p-values less than 0.001, ****p-values less than 0.00001. Data obtained on all the 25 geckos together independently of morphotype. See the Supplemental Information for differences among morphotypes.

also noteworthy that EE, which quantifies aspects of the shape of individual spots, is only weakly correlated with the other indices, with the exception of the mean elongation (EL), which also quantifies aspects of the shape of individual spots. This indicates that the spot shape only weakly depends on size or distribution of the spots. A negative correlation value indicates that two indices are anti-correlated. For example, fractional melanistic area (FM) and peak length (PL) are moderately anti-correlated since fractional melanistic area –the proportion of melanistic area to total skin area- tends to increase with the number of spots while peak length a measure of the typical distance between spots - decreases with the number of spots. EE and EED are moderately anti-correlated, which indicates that spots with large eccentricity tend to have a lower variation in their eccentricity, which indicates an eccentricity that is non-random.

We also computed the correlation coefficients of within-individual indices of the various body parts. The results are summarized in Fig. 6. A total of 127 out of the 294 correlation coefficients were statistically significant at the 0.05 level of significance (39.8%), meaning that we can statistically reject the hypothesis that variation of these indices is independent among these body parts. The number of statistically significant coefficients varied substantially by the pair of body parts. Correlation is significant for most indices for the two front leg patterns and the head and tail patterns (FL-FR, HD-TA; each 11 out of 14). To a somewhat lesser extent, the indices for the two back legs tended to be correlated (BL-BR; 9 out of 14 significant). Correlation between front and back legs (FL-BL, FL-BR, FR-BL, FR-BR) was much weaker. The trunk pattern was most strongly correlated with the tail pattern (TR-TA; 8 out of 14). Interestingly, within-individual correlation tends to be weak for indices describing the typical shape of the spot (EE, EL), whereas measures of the relative size of the spots (FM, SS, SA) tend to be highly correlated between body parts.

We performed a principal component analysis on the 14 indices taken across all the 25 studied geckos and across the different body parts, the results of which are summarized in Table SA3 and Figs. 7–9. A principal component analysis is a statistical method to convert a set of observations (in our case, among 14 indices that have many overlaps in the information they are describing) into a smaller number of uncorrelated variables called

|  | FM | SS | SSD | EE | EED | PL | MD | MDD | SA | SAD | SI | SID | EL | ELD |
|---|---|---|---|---|---|---|---|---|---|---|---|---|---|---|
| FL-FR | 0.95**** | 0.84*** | 0.48 | 0.4 | -0.26 | 0.78** | 0.89**** | 0.78** | 0.85**** | 0.86**** | 0.98**** | 0.56* | 0.7** | 0.74** |
| HD-TA | 0.7**** | 0.43* | 0.51** | 0.19 | 0.011 | 0.64*** | 0.72**** | 0.45* | 0.53** | 0.58** | 0.65*** | 0.5* | 0.53** | 0.4 |
| BL-BR | 0.98**** | 0.81*** | -0.09 | 0.45 | -0.36 | 0.43 | 0.85**** | 0.77*** | 0.91**** | 0.73** | 0.96**** | 0.79*** | 0.64* | 0.33 |
| FR-HD | 0.89**** | 0.57* | 0.45 | 0.017 | -0.0058 | 0.42 | 0.66** | 0.56* | 0.79*** | 0.72** | 0.84*** | 0.56* | 0.12 | 0.28 |
| TR-TA | 0.71*** | 0.61** | 0.45* | 0.15 | 0.27 | 0.17 | 0.34 | 0.34 | 0.67*** | 0.43* | 0.56** | 0.6** | 0.53** | 0.2 |
| BL-TR | 0.85**** | 0.61* | 0.05 | 0.29 | 0.34 | 0.29 | 0.61* | 0.22 | 0.6* | 0.42 | 0.84**** | 0.037 | 0.57* | 0.16 |
| HD-TR | 0.66*** | 0.41 | 0.16 | 0.67*** | 0.38 | 0.26 | 0.61** | 0.39 | 0.37 | 0.052 | 0.66*** | 0.32 | 0.57** | 0.45* |
| BL-HD | 0.52* | 0.43 | 0.23 | 0.18 | -0.15 | 0.12 | 0.83**** | 0.57* | 0.33 | 0.17 | 0.67** | 0.51* | 0.29 | -0.12 |
| BR-TR | 0.81*** | 0.54* | 0.17 | 0.043 | -0.14 | 0.2 | 0.4 | 0.24 | 0.64** | 0.62* | 0.73** | -0.063 | 0.3 | 0.2 |
| FL-HD | 0.81*** | 0.31 | 0.28 | -0.14 | -0.22 | 0.36 | 0.62* | 0.29 | 0.56* | 0.57* | 0.9**** | 0.17 | 0.19 | 0.37 |
| FL-TA | 0.65* | 0.48 | 0.62* | 0.44 | -0.18 | 0.4 | 0.64* | 0.56* | 0.45 | 0.46 | 0.67* | -0.096 | 0.19 | 0.34 |
| FR-TR | 0.61* | 0.25 | 0.4 | 0.54* | -0.00097 | 0.8*** | 0.63* | 0.28 | 0.37 | 0.33 | 0.66* | 0.13 | 0.16 | 0.2 |
| BL-FL | 0.61* | 0.56 | -0.21 | -0.2 | -0.43 | 0.064 | 0.56 | 0.67* | 0.68* | 0.2 | 0.84*** | -0.068 | -0.22 | -0.52 |
| BL-FR | 0.65* | 0.41 | 0.008 | 0.13 | 0.15 | 0.13 | 0.64* | 0.67* | 0.44 | 0.15 | 0.84*** | 0.083 | 0.014 | -0.46 |
| BL-TA | 0.71** | 0.54* | -0.023 | 0.3 | 0.21 | 0.31 | 0.47 | 0.71** | 0.52* | 0.35 | 0.45 | 0.34 | 0.16 | -0.4 |
| BR-FR | 0.62* | 0.59* | 0.11 | -0.3 | -0.36 | 0.066 | 0.44 | 0.18 | 0.62* | 0.25 | 0.84*** | 0.15 | -0.05 | -0.23 |
| BR-HD | 0.53* | 0.32 | -0.4 | 0.18 | 0.024 | 0.5 | 0.74** | 0.57* | 0.38 | -0.0023 | 0.61* | 0.3 | 0.24 | 0.18 |
| FR-TA | 0.69** | 0.39 | 0.14 | 0.22 | 0.21 | 0.19 | 0.64* | 0.8*** | 0.44 | 0.38 | 0.7** | 0.13 | 0.16 | 0.12 |
| BR-FL | 0.54 | 0.71** | 0.38 | -0.24 | 0.48 | 0.01 | 0.32 | 0.32 | 0.74** | 0.33 | 0.82** | 0.033 | -0.022 | -0.081 |
| BR-TA | 0.7** | 0.74** | 0.33 | 0.11 | 0.21 | 0.47 | 0.36 | 0.48 | 0.61* | 0.38 | 0.46 | 0.048 | 0.3 | 0.23 |
| FL-TR | 0.5 | 0.27 | 0.28 | -0.16 | -0.53 | 0.54* | 0.55* | 0.33 | 0.41 | 0.31 | 0.65* | 0.058 | 0.11 | 0.47 |

**Figure 6 Correlation coefficients of within-individual indices.** Values are color coded from negative (blue) to positive (red). Each row corresponds to a different pair of body parts indicated as in Fig. 2. *E.g.* "BL-HD" denotes comparison of the left back leg and the head of an individual. Columns correspond to the 14 indices of pattern characteristics as in Table 2. Correlation coefficients were calculated based on all geckos who had patterns on the corresponding body parts; *e.g.* the "BL-HD" values are computed over all geckos with patterned back legs and patterned heads. One star (*) indicates *p*-values less than 0.05, **p*-values less than 0.01, *** *p*-values less than 0.001, **** *p*-values less than 0.00001. Rows are sorted by the number of significant correlation coefficients. Data obtained on all the 25 geckos together independently of morphotype. See the Supplemental Information for differences among morphotypes.

the principal components. Together, the first seven components account for over 94% of the total variance between the 132 different patterns we analyzed. The first principal component (explaining 45% of the variance) has larger coefficients for those indices that are associated with the average size of the spots, while the second component (18% of the variance) tends to have larger coefficients for those indices associated with the characteristic wavelength (roughly corresponding to the distance among spots) of the pattern. The third principal component (12% of the variance) is strongly associated with EE and EED, which describes how close the individual spots are to circular shapes, and how much they vary in this regard.

Figures 7–9 summarize the PCA data graphically. Together PC1 and PC2 (generally, variation in spot size and separation, respectively) explain nearly two thirds of the pattern variation (62.8%) and provide insight into systematic differences among patterns of the legs, head, tail and trunk. Clusters of leg patterns are well separated and distinct from clusters of the head and tail patterns for plots of PC1 *versus* PC2 (Fig. 7). Since they sort along the PC1 axis (roughly, summarizing measures of spot size), we find that for example the head and tail patterns tend to have larger average spot areas, while leg spots have smaller average spot areas (Fig. 8, top right and bottom left). This is true for both the absolute size of the spots (indices SA and SS) as well as the size of the melanistic area as a fraction of total skin area (FM), indicating that leg spots tend to be disproportionately smaller than

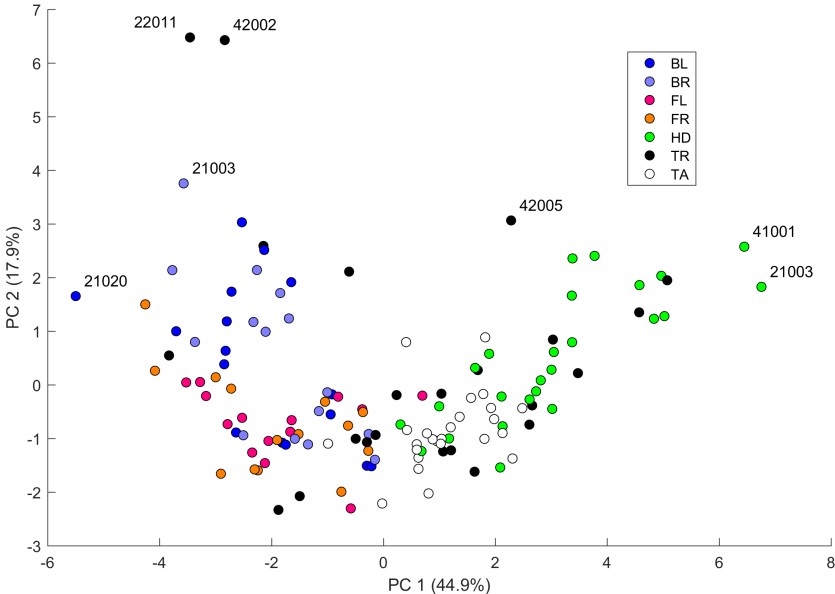

**Figure 7  Data visualization 1.** Plot of principal components 1 and 2. Each dot in the plot corresponds to a specific body part of a gecko. Body parts are colored coded and indicated by the same symbol as in Fig. 2. A few outliers are labeled *via* the corresponding gecko id (See Fig. A1 for images of all individuals. Note that the two outliers for the trunk patterns in the top left corner correspond to two normal morphs obtained from different sources, suggesting that grouping between these two individuals is not due to them being blood related). Data from all the 25 geckos with pattern in the specific body part indicated.

body spots. When looking at pattern variation only for legs, we observe that the front legs are clustered with lower PC2 values (PC2 is more associated with measures corresponding to the distance among spots) and also variation in the front legs tends to be smaller than variation in back legs (Fig. 9, left).

## Within-individual and between-individual distances in pattern space—a measure of the influence of developmental noise in pattern formation

### Pattern distances among pairs of legs

We investigated whether the patterns on each pair of the four legs on each gecko are more similar than the patterns on different geckos. In fact, variation in pattern among the two front legs or the two back legs for each gecko most likely reflects the level of developmental noise in pattern formation on the legs. In Fig. 9 (left panel)—where variation reflects mostly differences in the size of the spots and distance among them—a clustering of the legs of individual geckos is not readily visible, suggesting that the contribution of noise generally is more important than the contribution of genetic factors. It should be noted that although a considerable measurement error also contributes to the differences between the legs, random variation due to developmental noise is significantly larger than the measurement error (see Tables SA1 and SA2). We compared the within-individual differences in pairs of leg patterns to the between-individual differences in the same pairs (Fig. 10). We found qualitatively somewhat similar results for both the Mahalanobis distance (gray and black
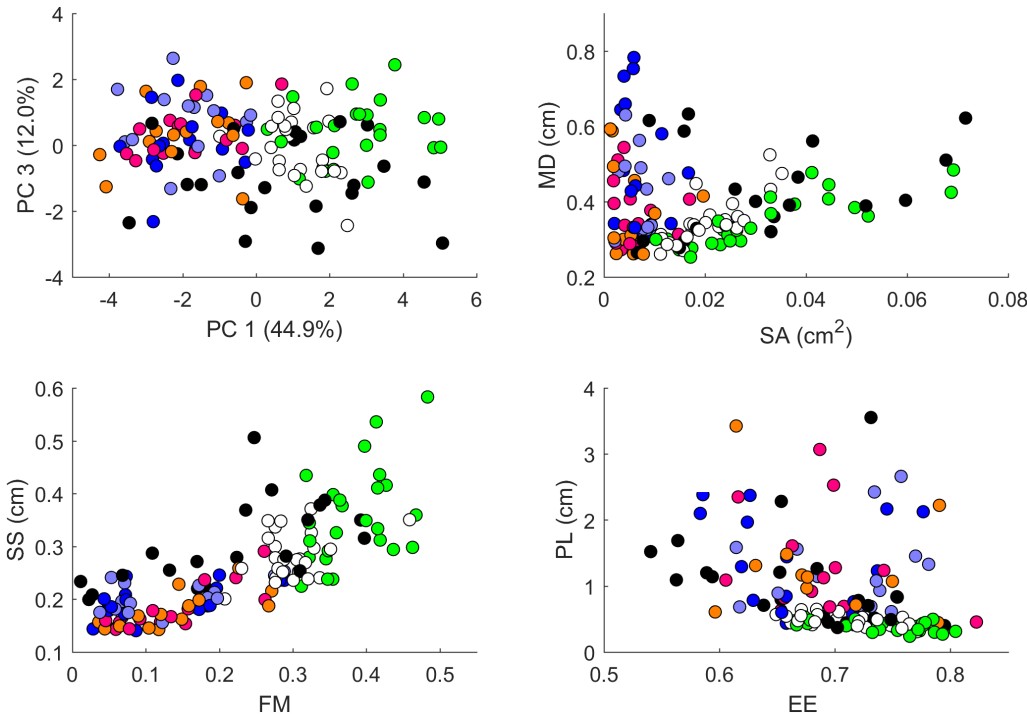

**Figure 8** **Data visualization 2.** Plots of principal components 1 and 3 (top left); spot area (SA) *vs.* mean distance between neighboring spots (MD; top right); fractional area (FM) *vs.* spot diameter (SS; bottom left) and ellipticity (EE) *vs.* peak length (PL; bottom right). Percentages in parentheses give the fraction of the total standard deviation explained by the given principal component. Body parts are color coded as indicated in Fig. 7. Axes bounds are chosen so that in some cases, a few outliers are not shown. Data from all the 25 geckos with pattern in the specific body part indicated.

bars in Fig. 10) and the Developmental Noise distance (pink and red bars in Fig. 10) concerning the distance between the two front legs and the distance between the two back legs. In both cases, the within-individual distances were smaller than the between-individual distances. Their ratio was between 0.55 and 0.57 for the Mahalanobis distance and even 0.16–0.19 for the developmental noise metric (a ratio of 1.0 would indicate no difference between the within-individual and the between-individual distances). The differences were highly statistically significant except for the case of back legs and the Mahalanobis distance. The within-individual pair of the two front leg patterns is thus found to be very significantly more similar than two leg patterns from different individuals. (Statistical significance was assessed *via* nonparametric permutation tests, see the Appendix for more details). Similarly, the within-individual distance between the two back leg patterns is also found to be significantly smaller than the mean between-individual leg distance. For both distances, a front leg and a back leg of the same gecko were closer to each other than a front leg and a back leg patterns randomly taken from two different geckos. However, overall, the mean distance between leg patterns of the same gecko is 64% of the mean distance between patterns on different geckos for the Developmental Noise distance and 91% for the Mahalanobis distance, a not statistically significant difference in the latter case. This

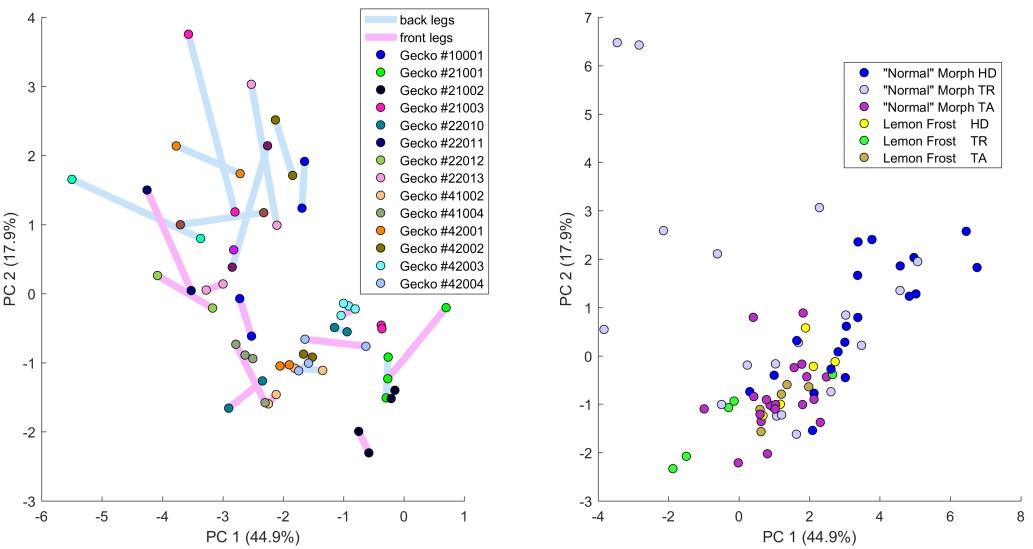

**Figure 9** **Data visualization 3.** Left: Plot of principal components 1 and 2 of all leg patterns. Geckos are color coded as indicated in the legend and each dot in the plot corresponds to a specific leg of a gecko. The two back legs of the same gecko are connected *via* a blue line; the two front legs by a purple ('fuchsia') line. Right: Plot of principal components 1 and 2 of head (HD), trunk (TR) and tail (TA) patterns. Geckos are color coded by morph.

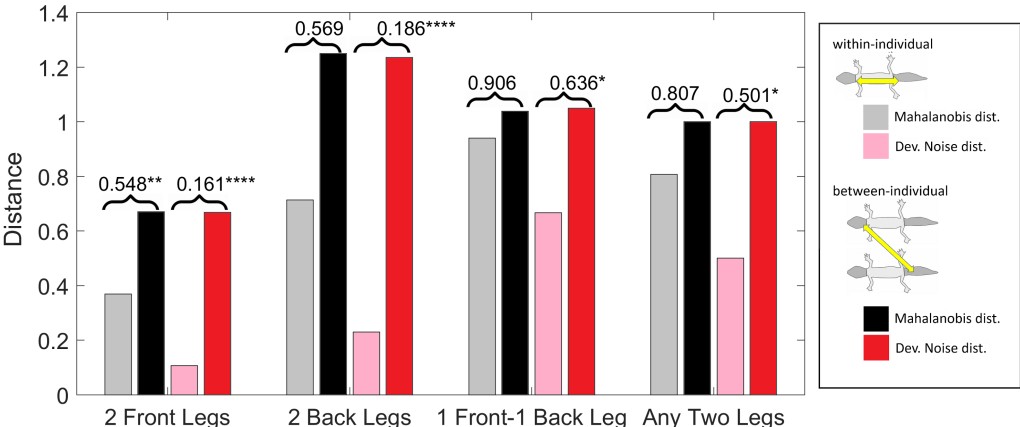

**Figure 10** **Distances of leg patterns.** (Squared) distances among leg patterns. For each type of leg comparison, within individual distance (gray and pink bars) and between individual distance (black and red bars) and their ratio (value indicated above each pair of bars) are shown. Gray and black bars show the Mahalanobis distance, pink and red bars show the Developmental Noise distance (see inset next to the figure). Squared distances are scaled so that the mean between-individual leg distance, *i.e.,* the distance between two leg patterns of different individuals, is 1 in each metric. Stars are based on the *p*-values for the null hypothesis that the mean within-individual distance is greater or equal to the between- individual distance; equivalently, that the ratio between the two is greater or equal to one. The alternative hypothesis is that the mean within-individual distance is strictly less than the between-individual distance. Data obtained on all the 25 geckos together independently of morphotype. One star (*) indicates *p*-values less than 0.05, ** *p*-values less than 0.01, *** *p*-values less than 0.001, **** *p*-values less than 0.00001.

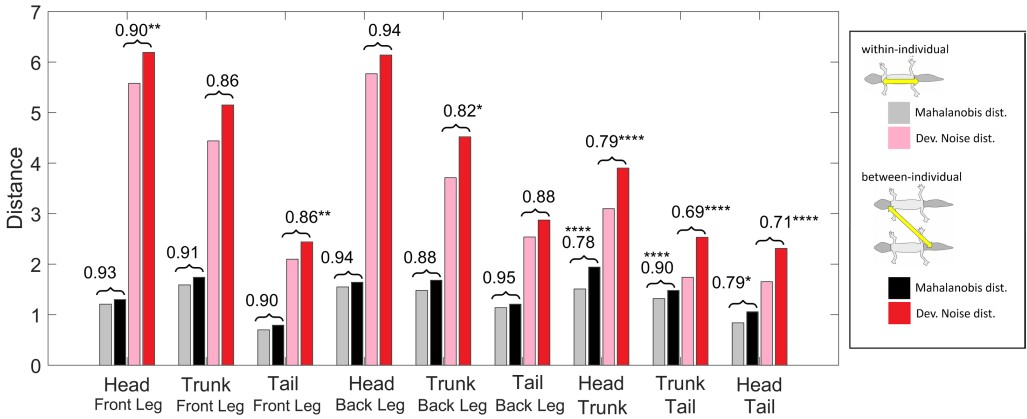

**Figure 11** **Distances of patterned body parts.** (Squared) distances among head, trunk, tail, and (front) legs. For each type of leg comparison, within individual distance (gray and pink) and between individual distance (black and red bars) and their ratio (value indicated above each pair of bars) are shown. Gray and black bars show the Mahalanobis distance, red and pink bars show the Developmental Noise distance (see also the legend to Fig. 10). Scaling of distances as in Fig. 10. Stars are based on the $p$-values for the null hypothesis that the mean within-individual distance is greater or equal to the between-individual distance; equivalently, that the ratio between the two is greater or equal to one. The alternative hypothesis is that the mean within-individual distance is strictly less than the between-individual distance. One star (*) indicates $p$-values less than 0.05, ** $p$-values less than 0.01, *** $p$-values less than 0.001, **** $p$-values less than 0.00001. Data obtained on all the 25 geckos together independently of morphotype.

further indicates that there is large variation in leg pattern (Figs. 7 and 9), and specifically that the variation observed between legs within each individual is only slightly less than the one observed among individuals, consistent with the small within-individual correlation coefficients observed for some indices (Fig. 6). Our results indicate that within each gecko, two front legs and the two back legs are much more similar to each other than a front and a back leg. This would suggest that the difference between a front and a back leg is not just due to developmental noise. In fact, the results support the hypothesis that the mechanisms - including timing of it - of pattern formation and/or regulation of pattern establishment are distinct for the front and back legs.

### Within-individual and between-individual pattern distances among heads, trunks, tails and legs

We also investigated the distances between the head, trunk, tail, and leg patterns. The results are summarized in Fig. 11. The distances are scaled with the same factor as in Fig. 10, *i.e.,* the average distance between the patterns of two legs from different geckos is scaled to 1. We compared the within-individual and between-individual distances for all possible pairs of head, trunk, tail and leg patterns.

The comparisons of leg patterns with other body parts (legs-head, legs-trunk, legs-tail) yielded that the mean within-individual distance was slightly smaller than the mean between-individual distances for both the Mahalanobis distance and the Developmental Noise distance. However, the differences were relatively small, between 5% and 12% for the Mahalanobis distance and 6% and 18% for the Developmental Noise metric. For
the Mahalanobis distance, these differences did not reach statistical significance. In other words, if you take a leg pattern of one individual, the difference between this leg pattern and, say, the head pattern of the same individual is on average not statistically significantly smaller than the difference to the head pattern of a second individual. This is different for the tail, head and trunk patterns. For all three pairs - trunk and tail; head and tail; head and trunk - the within-individual distance is significantly less than the between-individual distance. This means that on average, the patterns of any two body parts of the same animal, *e.g.*, the head and the trunk, is more similar than two patterns from different animals, *e.g.*, the head pattern of one animal and the trunk pattern of another animal. This holds true for both Mahalanobis and Developmental Noise distances.

## DISCUSSION AND CONCLUSIONS

The goal of this paper was to develop tools and methods to quantitatively study skin pattern variation within and between individuals. We developed a pipeline to collect the data from images of live geckos and then computed various geometric indices to describe characteristics of the patterns. Similar approaches have been taken for giraffe coat patterns (*Lee, Cavener & Bond, 2018*) and salmonid fish skin patterns (*Miyazawa, Kondo & Okamoto, 2010*); however in our work, we captured different aspects of pattern elements and looked at variation for each of the elements, and we also used two concepts of distances to quantify the degree of similarity of patterns as whole.

Our method is not only relevant for the analysis of experimental data, but also for the evaluation of mathematical models of skin pattern formation. There are in fact many such models (*e.g.*, *Murray, 2002*; *Cruywagen, Maini & Murray, 1992*; *Painter, 2001*; *Cooper et al., 2018*; *Kondo, Iwashita & Yamaguchi, 2009*), encoding various hypothesized mechanisms of pattern formation. It is far from straightforward to rigorously compare the synthetic patterns these models generate with the real skin patterns due to the complexity and irregularity of the patterns. Most authors typically either compare only one or two indices, such as the typical wavelength of the pattern, or just rely on the judgment of human pattern recognition. While this method of comparing patterns "by eye" lacks rigorous quantification, in some ways it is arguably far more developed and sophisticated than current methods of pattern comparison based on lower-dimensional, quantifiable measures. Therefore, our method represents a hybrid approach in which numerous, not necessarily independent measures (the 14 indices used in this work, considered as pattern elements) are chosen at the discretion of a human, based on the pattern variation and characteristics that are perceived as important. The values of these measures are then obtained using automatic methods and mathematical definitions are used to supply suitable weights for the measures to define the distances between patterns. This approach therefore permits us to fully depict variation in melanistic patterns within and among individuals and to quantify differences.

The 14 indices used for this project were chosen for the characteristics of the patterning we found among the 25 geckos, which were largely spotted but also occasionally included stripes and more complex shapes. Different pattern features and characteristics would
motivate different indices, which could then be analyzed using the methods described here. An exciting avenue for developing relevant indices for complex patterns (for example, for patterns at different points along the transition between spots and stripes) is to map patterns to a parameter space defined by a pattern generating algorithm such as a reaction diffusion model. In a previous work, we used statistical methods to map complex intermediate patterns to parameters corresponding to regions in a defined "LALI" pattern space (*Kiskowski et al., 2019*).

## Hierarchy of patterned body parts based on developmental sequence of melanistic patterning

We found an identifiable hierarchy of melanistic patterning head/ tail → trunk →legs in the studied leopard geckos; for example presence of patterns on the front legs also entails patterns on the trunk, tail and head, but not necessarily on the back legs (Fig. 3). There is no clear hierarchy between the front legs and the back legs, although for the "normal" geckos in our sample, unpatterned front legs implied unpatterned back legs. These results point to a corresponding order of the establishment of patterns during development: it appears that pattern formation occurs simultaneously in an anterior-posterior and a proximal-distal direction, forming first on the head, then on the trunk, followed by the legs. Patterning of the front legs and the back legs appears to be independent, due to the independent presence of pattern, the low correlation of the pattern indices (Fig. 6) and the fact that within-individual comparisons of front and back legs yielded very similar results as between-individual comparisons (Fig. 10). Pattern formation and establishment on the tail appears to be based on a related mechanism as the head and trunk, as indicated by the similarity of tail and head patterns, as well as tail and trunk patterns. The observed hierarchy of patterning partially follows pigmentation development in this species. Melanistic pigmentation in the leopard gecko starts to appear around the developmental stage 40 (hatching occurs at stage 42) as a banded pattern on the body and spots on the front legs (but not on the back legs) (*Wise, Vickaryous & Russell, 2009*). At the beginning of developmental stage 41, the banded pattern is clearly distinct across the body while a spotted pattern occurs on the upper part of the front leg. However, by the end of this stage, pigmentation is occurring across the whole body (*Wise, Vickaryous & Russell, 2009*). Although the body (head, trunk, and tail) of hatchling and juvenile leopard geckos generally presents a banded pattern, this species undergoes ontogenetic color and pattern changes, with adults generally having a spotted pattern (Fig. 1) (*Landová et al., 2013*). Ontogenetic change in color pattern however does not occur for the legs after hatching (see above). Therefore, head, tail, and trunk follow a process of pattern development and establishment that is different from the one occurring on the legs, with the pattern on the front legs establishing before the one of the back legs (*Wise, Vickaryous & Russell, 2009*), potentially explaining why absence of pattern is more common in the back legs than the front legs and in the front legs more than in the rest of the body.

## General pattern variation across geckos and body parts
### Variation and correlation among pattern indices

We found large variation and strong correlation among indices related to the amount of melanistic area and the density and size of the spots among the different studied geckos and among the different body parts (Figs. 5 and 6). Although variation in spot size across body parts and among individuals may partially be related to size differences, this is less the case for the fraction of melanistic area on the total area. Independently on the observed variation, once spots are formed, their average shape is similar across individuals (Table SA4), suggesting a strong constraint on this pattern element (EE), which is also weakly correlated to the other indices (Fig. 5). While variation in spot size and density has also been observed in other organisms (*e.g.*, *Asai et al., 1999*; *Morgan et al., 2014*; *Rudh, Rogell & Höglund, 2007*; *Balogová & Uhrin, 2015*; *Druml et al., 2017*), less is known about variation in spot shape. Potential genetic or developmental mechanisms may have evolved to ensure maintenance of spot shape and low variability of this trait. On the other hand, other elements of the spotted pattern (*e.g.*, density and size) may be freer to vary in a coordinated way –as suggested by the observed high positive correlation between some indices. Future research could further investigate if low variation in spot shape also occurs in other spotted vertebrates and if it is similarly achieved across organisms. In zebrafish, different alleles of the *leopard* gene result in changes in spot size, density, and connectivity among spots, suggesting that this gene may regulate the synthesis of an activator in a model of reaction–diffusion pattern formation (*Asai et al., 1999*). Later studies identified the role of *leopard* in regulating interaction among melanophores (or among xanthophores) and in controlling boundary shape for the spots (reviewed in *Kondo, Iwashita & Yamaguchi, 2009*; *Singh & Nüsslein-Volhard, 2015*). Similarly, in horses, two genes with different alleles determine the occurrence and amount of melanistic spots on a white colored coat (*Druml et al., 2017*). The availability of the leopard gecko genome (*Xiong et al., 2016*), the relative easiness to breed this species, and the existence of CRISPR-Cas9 technology already tested to create mutations in lizards (*Rasys et al., 2019*) will allow us to develop future research to uncover the genetic basis of variation in pattern elements in this species, similarly to what has been done for mammals and other non-mammalian model species (*Van Belleghem et al., 2020*; *Concha et al., 2019*).

### Variation and correlation in pattern among body parts

Phenotypic correlation among traits, in this case the correlation of patterns among different body parts of the same individual, may provide information on how these patterns are related developmentally. Phenotypic correlation was investigated in two ways. The first is the standard method of Pearson correlation coefficients for each pair of body parts and for each measurement, summarized in Fig. 6. The second measure is the ratio of the mean within-individual distance and the mean between-individual distance in pattern space for each pair of body parts, summarized in Figs. 10 and 11. Patterns on the legs are statistically almost independent of patterning on the head, trunk and tail. In contrast, the similarity of head, trunk and tail patterns, as well as the similarity of the two front legs and the two back legs for the same animal are statistically significant for both metrics.

The similarity of pattern variation observed on the head, trunk and tail suggests that patterning mechanisms are most likely not independent among these body parts, and the same holds for the two front legs and the two back legs (Figs. 10 and 11; see also Fig. 9 for each index separately). However, as melanistic patterns in the legs, and especially the back legs, are more variable and independent in their variation from the rest of the body, this may indicate a different timing or developmental mechanisms of pattern formation and establishment in these body parts. In this sense, the relatively easiness of captive-breeding of this species may provide a unique opportunity into understanding the underlying genetic and developmental processes and mechanisms producing the observed variation in color pattern in the different body parts (for similar questions, see *Cieslak et al., 2011*; *Druml et al., 2017*; *Wasik et al., 2014*).

## Comparison of within-individual and between-individual differences in leg patterns as a measure of developmental noise

A within-individual comparison of the two front leg patterns yields that the front legs of an individual gecko are significantly more similar than two between-individual front leg patterns. The same holds true for the two back legs. A simple measure of the magnitude of the contribution of developmental noise is given by the ratio of the mean within-individual distance and the mean between-individual distance, which is the amount of variation presumably due to developmental noise alone normalized by the average amount of pattern variation due to all sources (including genetic and environmental). A ratio of 0 would indicate that the legs of individuals show no variation at all within a gecko, meaning that developmental noise plays no part in the establishment of patterns at all. Conversely, a ratio of 1 would mean that the variation between leg patterning of the same animal is indistinguishable to the variation between patterning for two different animals. This would indicate that the process of patterning even between geckos would be entirely dominated by random noise. In our data, the contribution of developmental noise to patterning is quite large by this measure with ratios of within-individual distances to between-individual distances between 0.55 and 0.16 for the Mahalanobis and Developmental Noise metrics, respectively, for the front legs and 0.57 and 0.16 for the back legs. While the measurement error contributes to this estimate—it accounts for between 47% and 33% of the within-individual distances (Table SA1)—the variation due to developmental noise exceeds this error significantly (Tables SA1 and SA2). These indices indicate that although variation in melanistic pattern observed within individuals is not produced by developmental noise alone, overall, developmental noise has a very strong influence on this variation. Together with controlled captive-breeding experiments, the combination of mathematical modeling (see section below) and empirical data can be used in the future to further investigate the relative importance of genotype, environment and developmental noise on the variation in color pattern on the different body parts in these animals. Furthermore, our methodological approach can also be applied to other patterned organisms to study similar questions.

## Methodological significance for the analysis of mathematical models

Our method also serves the purpose of establishing a systematic high dimensional quantitative approach to the analysis of synthetic patterns produced by mathematical

models of skin pattern formation (see *e.g.*, *Murray, 2002*; *Cruywagen, Maini & Murray, 1992*; *Painter, 2001*; *Cooper et al., 2018*; *Kondo, Iwashita & Yamaguchi, 2009*). Indeed, the distance between synthetic patterns and the actual patterns is a quantifiable overall measure of how similar the synthetic patterns produced by such models are to the actual patterns. More importantly, our method gives a way to quantify the effect of developmental noise on patterns, which in turn can be used to calibrate and test mathematical models of skin pattern formation. More concretely, we can think of the two front leg patterns of an animal as two points in our 14-dimensional pattern space. The two leg patterns are similar, but not identical. These two points represent the variation from a "typical" pattern that corresponds to the environmental and genetic conditions of the particular individual's development. We postulate the differences between the two legs to be the effect of developmental noise. Indeed, we can conceptualize abstract sets of possible patterns that can be attained under the same combination of environmental and genetic conditions, but with the random contribution of developmental noise. In a previous work, we have called this set a 'phenotype cloud' (*Kiskowski et al., 2019*); see Fig. 12 below. In our case, it is a subset of a 14-dimensional pattern space. These phenotype clouds can be interpreted as confidence regions for the phenotypes of individuals with the same genetics and similar environments, extending the ideas of confidence intervals to higher dimensions. One can think for example of a 95% phenotype cloud as an abstract set of patterns that contains a randomly generated pattern (with fixed environmental and genetic conditions, but a random contribution of developmental noise) in 95% of the cases. Since the distribution function for random noise is completely unknown, and we only have two sample points for each instance—the pairs of front legs or the pairs of back legs for each individual—, the actual shapes of these phenotype clouds are unclear. However, in the context of stochastic mathematical models of pattern formation, such model-dependent phenotype clouds can be determined computationally (*Kiskowski et al., 2019*), which then allows us to test such model prediction against the empirical data.

Thus our approaches can be used to test the validity of mathematical models for skin patterning, and gain insights and formulate predictions on the cellular and genetic mechanisms of pattern formation (*e.g.*, *Maini, 2004*; *Othmer et al., 2009*). Besides providing a framework for quantitatively analyzing various aspects of patterns, the concept of the phenotype cloud gives an additional empirical approach to interrogating models.

## ACKNOWLEDGEMENTS

We thank Nathan Katlein for his initial help in developing the method for obtaining the gecko images. We are thankful to Gopal Murali, William Allen, and Julien Claude for discussing the correlation of color pattern among distinct body parts in animals during the early phases of writing this article. Julian Claude also provided helpful comments to improve this article. We also thank Ekkehard Glimm for very helpful discussions about the statistical analysis, in particular for discussions about computations of $p$-values. We are thankful to Tony Gamble, Aaron Griffing, and John Scarbrough of Geckoboa for discussion about color pattern and color pattern selection in the pet trade for the leopard gecko and
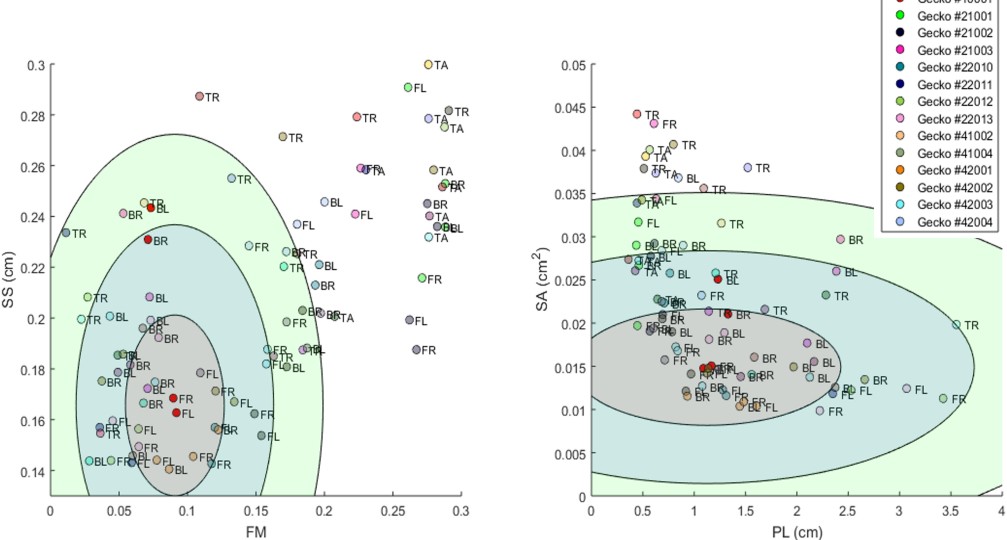

**Figure 12 Visualization of phenotype cloud.** Illustration of the concept of a "phenotype cloud". The images show the gecko pattern data in FM-SS space (left) and PL-SA space (right). Each gecko corresponds to a unique color and the body part is indicated by a two letter-combination (see Fig. 2). The ellipses shown are visualizations of the concept of phenotype clouds of gecko #10001 (indicated in bright red). In fact, these ellipses are projections of certain balls (with respect to the Developmental Noise metric) in 14-dimensional pattern space onto the FM-SS subspace (left) and on the SA-PL subspace (right). These balls are centered at the centroid of the two front leg patterns of gecko #10001 (indicated in bright red). The radius of the innermost ball is equal to the distance of one of the front legs to the centroid in the 14-dimensional pattern space. The other concentric balls have twice and three times this radius. Phenotype clouds can be conceptualized as such balls, *e.g.*, the 95% phenotype cloud is a set that contains randomly generated patterns with the same environmental and genetic conditions and the same level of developmental noise as the front legs of gecko #10001 in 95% of the cases. While it is not possible to generate such phenotype clouds for a single pattern from our data empirically—we only have two data points for each cloud, corresponding to the left and right leg—it is possible for stochastic mathematical models of pattern formation. The size of the cloud then indicates the contribution of developmental noise to pattern formation - the larger the cloud, the larger the influence of developmental noise.

to Matt Vickaryous for discussion about melanistic color pattern formation, especially in regenerated tissues. We are thankful to Juan Daza and to an anonymous reviewer for helpful comments on an earlier version of this article.

### Funding
The authors received no funding for this work.

### Competing Interests
The authors declare there are no competing interests.

## Author Contributions

- Tilmann Glimm and Maria Kiskowski conceived and designed the experiments, analyzed the data, prepared figures and/or tables, authored or reviewed drafts of the paper, and approved the final draft.
- Nickolas Moreno performed the experiments, authored or reviewed drafts of the paper, and approved the final draft.
- Ylenia Chiari conceived and designed the experiments, performed the experiments, analyzed the data, prepared figures and/or tables, authored or reviewed drafts of the paper, and approved the final draft.

## Animal Ethics

The following information was supplied relating to ethical approvals (i.e., approving body and any reference numbers):

All experiments were carried out in accordance with George Mason University animal use (IACUC) protocol # 1430668.

## Data Availability

The images of the gecko body parts which have been manually isolated from photographs taken in the lab, extracted data (as CSV tables) and MATLAB code are available at OSF: Glimm, Tilmann, and Maria Kiskowski. 2021. ''Gecko Patterns Data and Code.'' OSF. August 9. osf.io/zauwe.

## Supplemental Information

Supplemental information for this article can be found online at http://dx.doi.org/10.7717/peerj.11829#supplemental-information.

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
