# Peer review of "Capturing and analyzing pattern diversity: an example using the melanistic spotted patterns of leopard geckos"

_PeerJ, doi:10.7717/peerj.11829_

## Round 0.1 · original submission · Minor Revisions

Both reviewers found your study very valuable. I do not see the need to itemize my perspective for you as it agrees with all that posited by them. Please, follow carefully their suggestions, which I find highly compelling.

·

Basic reporting

Capturing and analyzing pattern diversity: an example using the melanistic spotted patterns of leopard geckos by Glimm et al.

I found this project very interesting and although I must confess that I cannot comment on the methods, because I don't have the computational background, looks like the automatization and capture methods are adequate. I have made a few comments directly on the PDF, but I would mention and explain these here. I also give away my anonymity statutes and the authors can contact me directly if needed. The paper really needs a throughout review of the format, the citations are inconsistent, sometimes et al., is written without the period or without the comma.

Juan Diego Daza (juand.daza@gmail.com)

Experimental design

Regarding the seven regions delimited, some regions seems a little bit arbitrary. I also noticed that the manus and pes were not included, despite the fact that they exhibit important variation. Portions of the cervical and sacral regions not included, but I guess this was done to avoid problems causes by gradual transition from one pattern to the other in adjacent regions. This is not necessary wrong, but it should be stated (I am sorry if you already did this). I also noticed that the sample of 25 geckos, although includes a good representation of the morphs available, it is not complete. I wonder if it would be worth including the two melanistic extremes (even from the theoretical point of view); Leopard geckos are know to have complete melanistic and albino morphs. This is probably not needed to include, but then again, at least mention this on the background information. I don't think this last point is critical, cause I don't expect a combination of these extreme forms with he morphs identified (e.g. a gecko with a white head and pigmented trunk).
Finally, this could be a matter of semantics, but melanistic while referring to black pigments, technically won't be classified as color, but as shade. Black since it doesn't reflect light, is not a color.

Validity of the findings

I think the paper has very interesting findings, especially the idea that different modules in the gecko body are independent. By the way, makes me wonder why the term module is not use?

Additional comments

I find this study very innovative for the study of gecko melanism. I think the variation observed can be expanded, especially if including the extremes is a possibility.

Reviewer 2 ·

Basic reporting

Tilmann Glimm and colleagues present in this manuscript a comprehensive strategy to quantify pattern variation. Their strategy uses 14 indices that capture characteristics of spotted patterns. These indices can then be used to compare patterns on body parts or between individuals using PCA type of analyses. Within the PCA space they also provide two normalized distance measures that allow to statistically compare patterns. They us this strategy on a set of gecko images and obtain novel insights into their pattern development. They also, interestingly, note that their strategy can be used to compare mathematical models of pattern development with observational data.

I read this manuscript with much interest and I found that the complex concepts are incredibly clearly explained. I think this study will provide a valuable benchmark for other researchers interested in studying pattern variation, which to me is notoriously difficult to quantify. The strategies to cope with sampling noise, developmental noise and individual differences seems unique and well thought out. I have only small suggestions and questions that I hope the authors could maybe elaborate a bit on.

Experimental design

no comment

Validity of the findings

no comment

Additional comments

Maybe one larger comment that I think would benefit the community is to make sure the pipeline is easily accessible and well annotated. I am not a Matlab user myself, because it is not a free software, so if I would want to use your scripts, I would have to follow a steep learning curve.

L58: These studies use landmarks to align images by transforming them to the same size and shape. The color patterns are then compared pixel by pixel, assuming that each pixels now presents a homologous feature.

L64: missing period after parentheses.

L68-79: I strongly agree. I wonder if saliency maps which should have the power to highlight important features in the images will provide possibilities for quantitative pattern recognition. https://arxiv.org/pdf/1312.6034.pdf

Also, distance scores can still be obtained from the machine learning algorithm, which provides a quantitative measure of similarity: see for example, https://advances.sciencemag.org/content/5/8/eaaw4967

L192-205: I am hoping the authors could elaborate on how this trial and error strategy would work for other studies with different organisms. Is there potential for bias, is image standardization necessary, …?

L211-227: I think I understand how this works for melanistic spots, but what to do with other color patterns (e.g. yellow or red spots) in other organisms? What would be the advantage over other automated pixel classification methods like k-means clustering or watershed algorithm.

L213: How does the green channel indicate intensity, rather than R or B? Could a HSV (hue, saturation, value) color space be more appropriate?

L216-217: instead of ‘determining by eye’, is it possible to get histogram distributions of RGB values and see where the threshold falls that separates RGB values of spots from background?

L233: How does the 350 pixels compare to the resolution of the images?

L254-260:

Can the overall absence of pattern also be included as an index?

All indices seem to focus on ‘spots’. How well would they perform on complex patterns of stripes or curves?

L563: What are these stages, days?

L605: Yes. https://www.sciencedirect.com/science/article/pii/S0960982219313168 and https://royalsocietypublishing.org/doi/10.1098/rspb.2020.1267

L664: duplicated period

The reference manager is mistakenly not recognizing ‘van’ as the first part of a last name:

L720: Van Belleghem S.M.

L724: van den Berg C.P.

---

## Round 0.2 · Minor Revisions

I think that you did a wonderful job in considering the reviewer's suggestions. However, a little point still needs your attention. In the introduction section, you begin writing on the discovery of color patterns and then simply speak on the melanistic patterns. After reading the response to our reviewers I quite understand your motives but maybe they are not so clear to our readers. Please incorporate a paragraph linking these two concepts.

In line 310 of the file with tracked changes, some problems with the writing appear.

---

## Round 0.3 · accepted · Accept

Thank you for following my suggestions. We can move on, nice paper!